# Moirai: A No-Code Virtual Serious Game Authoring Platform

**Andrei Torres [1,*]** , **Bill Kapralos [1,*]** , **Celina Da Silva [2]**, **Eva Peisachovich [2]** and **Adam Dubrowski [1]**

[1]    maxSIMhealth Group, Ontario Tech University, 2000 Simcoe Street North, Oshawa, ON L1H 7K4, Canada
[2]    Faculty of Health, York University, 4700 Keele Street, Toronto, ON M3J 1P3, Canada
*    Correspondence: andrei.torres@ontariotechu.net (A.T.); bill.kapralos@ontariotechu.ca (B.K.)

**Abstract:** Serious games, that is, games whose primary purpose is education and training, are gaining widespread popularity in higher education contexts and have been associated with increased learner memory retention, engagement, and motivation even among learners with special needs. Despite these benefits, serious games have fixed scenarios that cannot be easily modified, leading to predictable and dull experiences that can reduce user engagement. Therefore, there is a demand for tools that allow educators to create new modifications and customize serious game scenarios, and avoid the fixed-scenario problem and a one-size-fits-all approach. Here, we present and detail our novel virtual serious games authoring platform called *Moirai*, which uses a no-code approach to allow educators who may have limited (or no) prior programming experience to use a diagram-based interface to author and customize serious games focused on decision and communication skills development. We describe two case studies, each of which involved creating serious games for nursing education (one for mental health education and the other for internationally educated nurses). The usability of both games was qualitatively evaluated using the system usability scale (SUS) questionnaire and achieved above-average usability scores.

**Keywords:** serious games; virtual simulation; scenario editor; healthcare education

## 1. Introduction

When COVID-19 was declared a pandemic by the World Health Organization on 11 March 2020, governments worldwide issued lockdowns and stay-at-home orders that led to the shutdown of businesses and educational institutions. During the first week of April 2020, it was estimated that 195 countries had implemented school closures, affecting over 91% of the global student population [1]. The pandemic resulted in an abrupt move from traditional face-to-face instruction to remote learning. Although many educational institutions were gradually creating an online learning presence prior to the pandemic, the abrupt move to remote learning created numerous challenges and opportunities. Even before COVID-19, remote learning was on the rise for the last decade, with a forecast for the online education market at USD 350 billion by 2025 (not adjusted to consider the COVID-19 impact) [2]. Numerous online learning platforms such as Udemy, Linkedin Learning (formally called Lynda), and Coursera serve millions of users, and universities such as the University of Toronto and Stanford University are providing access to complete online courses. Technology is instrumental to online learning and can provide many benefits when implemented in the educational realm, including increased student engagement, access to the most updated information, sharing content, and interaction with others worldwide [3].

Concerning the students, the shift to remote learning during the pandemic has created a loss of in-person collaborative experiences, real-time feedback, and interactive discussions in the classroom. These losses can be a considerable disadvantage to healthcare education [4]. Furthermore, this shift has also resulted in isolation which has taken a noticeable toll on students' mental health [5].

Within the domain of healthcare education (in which our case studies presented in Section 4 are focused on), it has been suggested that COVID-19 may forever change how

future healthcare professionals are educated [6], and this may include a continuation of remote learning beyond the pandemic. Similar to in-class learning, the physical presence of trainees for inpatient (i.e., providing care for patients who are required to stay overnight) and outpatient (i.e., providing care that does not require overnight hospitalization) settings have been an integral part of early healthcare professions education health experiences since it provided learners with hands-on experience with actual patients. The COVID-19 pandemic has also greatly affected residency (training which leads to specialty or sub-specialty certification) and clerkship (supervised learning experience in a clinical setting) training. More specifically, clinic and hospital outpatient volumes have been drastically reduced as many non-urgent outpatient appointments have been cancelled, and inpatient hospital services have been reduced [7]. Additionally, some healthcare education fields were lacking or in decline even before the impact caused by COVID-19, such as mental health training [8] and education for internationally trained nurses (IENs) [9]. In the case of mental health training, the lack of mental health courses and nursing placements in mental health settings have made it difficult for learners to acquire the necessary skill set to care for persons living with mental illness competently. Furthermore, in the case of education for IENs, high-income countries (such as Canada) have been trying to remedy the shortage of nursing professionals by hiring IENs. However, many fail to pass the required standardized examination partly due to the lack of open access to educational resources, which should accommodate the work-life of IENs.

Aside from facilitating remote synchronous and asynchronous learning, technology has also been used to facilitate experiential ("hands-on") learning during the COVID-19 pandemic, as healthcare laboratories shut down and healthcare placements and clerkships were not available. Simulation provides a viable and safe alternative to real-world practice, allowing learners to train until they achieve a specific competency level [10]. Immersive virtual learning environments (iVLEs), including serious games, that is, video games applied specifically to learning and training, have been developed to provide effective educational tools for training across a wide variety of fields, in an engaging, safe, and cost-effective manner [11]. Serious games have been used to train healthcare professionals in various specialties, such as neurology, emergency, physiology, forensics, and urology, for both knowledge transmission or skill improvement [12]. This work refers to serious games as part of the virtual simulation modality within the iVLE domain (see Section 2.2).

Technology and iVLEs in the form of serious games have and will play a vital role in health education, particularly by facilitating remote learning during the COVID-19 pandemic and beyond. These pedagogical strategies offer an engaging alternative to classroom lectures and hands-on learning obtained during in-class laboratory sessions, clerkships, and residencies. Despite the benefits of iVLEs and, in particular, serious games, several limitations can restrict their use. More specifically, most serious games have fixed scenarios that cannot be easily modified, leading to predictable, disengaging, and repetitive experiences, which, after several sessions can negatively impact their effectiveness as teaching tools [13]. Therefore, there is a demand for novel tools that allow educators to easily and rapidly modify and customize the educational content, avoiding both the fixed scenario problem and a one-size-fits-all approach. Such an approach requires new development methods that allow educators who may have limited programming experience (if any) and technical knowledge to create/modify content in a simple manner [14]. Educators have expressed ambivalence about prescriptive digital tools compared to those that are modifiable [15], which supports our prior work demonstrating the acceptability of serious gaming scenario creators/editors in health education [16].

To overcome this one-size-fits-all approach, since 2019, we have been continuously developing a virtual serious game authoring platform called *Moirai* that allows educators of any domain to produce serious games. Our *Moirai* includes a user-friendly scenario editor which allows educators (who may have limited computer science/programming expertise) to easily create and modify customizable dialogue-based serious games, avoiding a one-size-fits-all, static scenario approach. The output is a self-contained serious game

that can be played using regular web browsers, and it can be imported by learning management systems (LMS) that support the Shareable Content Object Reference Model standard (SCORM). Our platform has so far been applied in the context of mental health/psychiatry education, cultural competency training, and interprofessional education for medical laboratory learners. However, it can be applied to any setting involving dialogue between two or more parties. Currently, adding or modifying 3D environments and characters needs to be implemented by game developers, but empowering educators and content creators to independently handle this is in our plans for future work. In this paper, we introduce and detail the *Moirai* and provide an overview of two case studies of serious games [17,18].

The remainder of the paper is organized as follows. Background information that includes an overview of commonly used terminology in the field of immersive virtual learning environments (iVLEs) and an overview of related work is provided in Section 2. A detailed description of the *Moirai* and the development methodology followed is provided in Section 3. Two case studies, with their specific contexts, scenarios and user studies, are presented in Section 4. A description of the challenges and current limitations of *Moirai* is provided in Section 5. Lastly, conclusions and development plans are provided in Section 6.

## 2. Background and Related Work

### 2.1. Serious Games

Games (and its more recent relative, computer games) have been created as leisure activities with the primary purpose of providing entertainment [19] and have grown into a multi-billion dollar industry. It is estimated that globally, the video games industry alone has generated USD 180.3 billion in revenue in 2021 [20], partially propelled by the COVID-19 pandemic, making it bigger than the US box office and North American sports industries combined (which were also impacted by the pandemic) [21,22]. However, not all games have the same goal. Serious games are designed for purposes other than, or in addition to, pure entertainment [23]. Serious games are used for various domains and purposes, such as health (e.g., surgical trainers) and the military (e.g., recruitment). Within serious games, there is also a subset called "games for learning", designed specifically with some learning goals in mind [23]. Furthermore, within "games for learning" we can also identify "educational games", which implies games used in formal educational settings [23]. Therefore, all "games for learning" are serious games, and all serious games are games.

A recent meta-analysis confirmed that serious games foster positive learning attitudes and enhance cognitive abilities and engagement [24]. Other studies show that serious games increase learner memory retention, engagement, and motivation in learning [25], even among special-needs learners [25]. Serious games are problem-solving approaches aligned with the social constructivist theory of learning [26]. Constructivism underlines the idea that knowledge is not transmitted to the student but is constructed through activity, such as participating in problem-based learning workshops, playing serious games, or interacting socially [26,27].

### 2.2. Simulation

Serious games are often confused with "simulators", "training simulators", and "simulation games" [28], and terms such as "simulation" and "virtual simulation" which, although broadly used, have meanings that depend on the context and domain. Although simulation games involve the simulation of realistic situations, it does not imply that they are serious games unless they have an underlying educational purpose. For example, "virtual simulation" can be used to describe physical simulation modalities (e.g., manikins) that are facilitated virtually (e.g., by web conference). However, it can also describe an in-person simulation conducted in a CAVE-like environment with images projected on the walls to help increase the participants' immersion. Additionally, it can also describe simulation in which the modality itself is virtual, with images and interactions generated by computer software [29]. Figure 1 presents a taxonomic tree depicting "virtual simulation" activities, and this work focuses on the "screen-based simulation" modality.

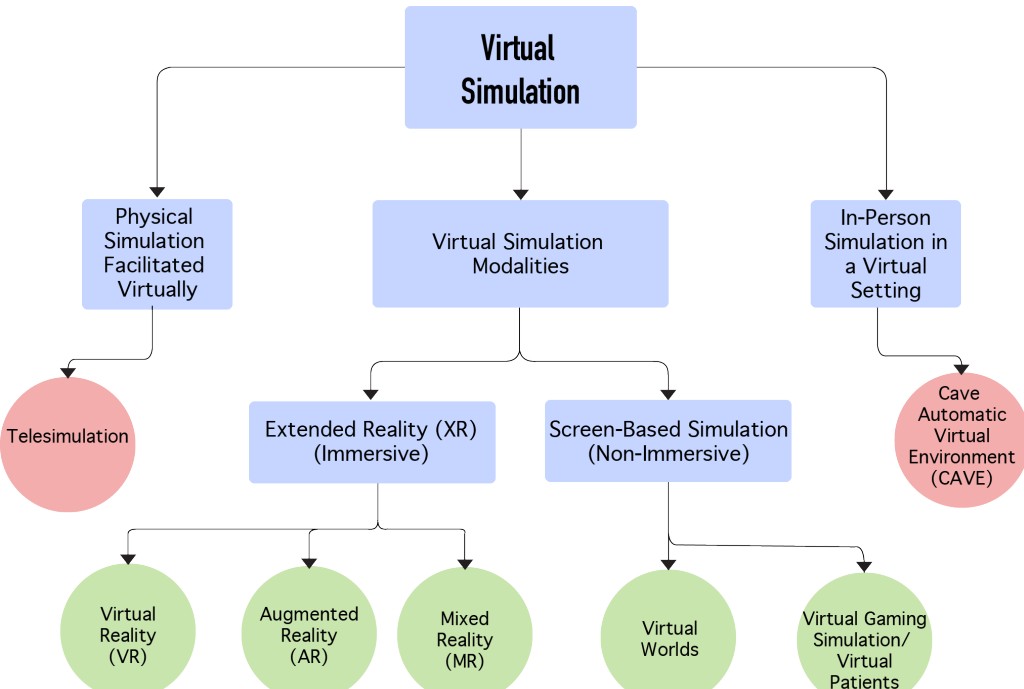

**Figure 1.** Taxonomy of Virtual Simulation. Reprinted from Ref. [29]. Available under the Creative Commons Attribution-NonCommercial 4.0 International License.

It is important to reinforce that "simulation" is a technique, not a technology, and it can replace or amplify real experiences with guided experiences that evoke or replicate substantial aspects of the real world in a fully interactive manner [30]. Additionally, simulations can employ different modalities, such as task trainers, standardized patients, computerized manikins, virtual reality, case studies, or hybrid, among other possibilities [31]. Regardless of the technology and modality adopted, the simulation will have a particular structure or architecture, a specific way that it unfolds and that the learner interacts with it. Verkuyl et al. propose the following classification [29]:

**Linear**
> The simulation follows a single navigation path, from node to node. Each node represents a step or point in the simulation, that may contain information, media (images or video), self-assessment questions, decision options, or other interactions. This presents limited interaction to the learner, as choices they make do not affect how the scenario unfolds, but they help keep the learner on course and are simplest to create.

**Branched**
> Again the simulation progresses from node to node, but at each decision point, the scenario will progress differently based on the choice made by the learner. At each decision point, typically a limited number of options (2–4) are presented to the learner. The ultimate outcome of the simulation depends on the learner's choices.

**Pseudo-branched**
> Similar to branched cases, but authored in a way to limit the number of possible paths and outcomes of the simulation. Some choices may bring the learner back to a previous node to choose another option, or some paths may merge back together.

**Exploratory**
> These simulations are open-ended, allowing the learner to explore a situation and decide what information to gather. For example, these take the form of a meeting between the professional and a client where the learner can choose what questions to ask, what assessments or investigations to perform, and other information to gather.

Once the learner is ready, they are asked to commit to an assessment of the situation or an action plan. Learners receive feedback on information gathered, information omitted, their efficiency, and their ultimate decision.

**Responsive**

Responsive virtual simulations are also open-ended, but powered by an engine (e.g., in healthcare it would be a physiology engine), so learners can choose assessments and interventions, and the simulator will respond in real-time to their actions. In healthcare, these are typically used for emergency or critical care simulations, where interventions such as intravenous fluids or endotracheal intubation have rapid effects.

**Communication**

Some platforms focus on re-creating communication between a service provider and a client, or among providers. They use natural language processing and artificial intelligence to interpret questions asked by the learner and then to provide appropriate responses. Other platforms focus on the nuances of how to phrase communications and are built more like a branching case, but with a relationship and trust with the client that can be built or lost.

**Procedural**

These virtual simulations are designed specifically to teach about a manual procedure or other step-wise processes. The flow of a scenario may resemble a linear or pseudo-branched case, but the focus will be on the procedure.

*2.3. Visual Programming*

Large technology companies such as IBM (IBM Watson Orchestrate: https://www.ibm.com/products/watson-orchestrate, accessed on 6 October 2022), Microsoft (Microsoft Power Apps: https://powerapps.microsoft.com/, accessed on 6 October 2022) and Amazon (Amazon Honeycode: https://www.honeycode.aws/, accessed on 6 October 2022) have been investing in empowering end users to create applications for their platforms [32], applying either a "low-code" or "no-code" approach. Low-code is a rapid application development (RAD) approach to software development that enables faster delivery of applications through minimal hand-coding and with visual building blocks such as drag-an-drop-drop and pull-down menu interfaces [33,34]. No-code is another RAD approach, often considered a subset of low-code. Although primarily visual, low-code still allows the developer to use scripts or some form of manual coding. In contrast, no-code takes an entirely hands-off approach, relying solely on visual tools [34].

Visual programming has been successfully used to teach novices basic programming concepts, and now it is being adopted beyond the educational domain by end users to develop applications tailored to their needs [32]. End users know their domain, needs, and specificities better than anyone else, and end-user development (EUD) has emerged as a field that caters to that audience, developing tools and activities to allow those who are not professional developers to create applications [35].

Forrester Research predicts that by the end of 2022, the low-code year-over-year growth rate will reach 40%, with a spending forecast to reach USD 21.2 billion by 2022 [36]. In addition, Gartner predicts that by 2024, 80% of technology products and services will be built by those who are not technology professionals, driven by the adoption of low-code and AI-powered tools [37].

A systematic review by Kuhail et al. [32] identified the following approaches to presenting visual programming tools:

**Form-based**

Allows users to construct a functional user interface (UI) by dragging and dropping visual components into a form.

**Diagram-based**

Allows users to construct a program by connecting visual components where the output of a component serves as data input to another component.

**Block-based**

Allows users to construct a program by combining visual blocks that fit together like a jigsaw puzzle.

**Icon-based**

Allows users to construct a program by connecting icons to represent data flow.

*2.4. Related Work*

Although serious games have been widely used in different fields and domains [38], including education [39–41], healthcare personnel training [42–44], software development education [45,46], and business education [47–49], they are generally created with a specific (rigid) goal, purpose, or scenario. The development of games, including serious games, is a very laborious and demanding effort in terms of time and cost. Therefore, instead of creating (and recreating) serious games from scratch, some authors propose frameworks and platforms to allow content creators to develop serious games that best fit their needs quickly.

Boada et al. [50] developed TAECon, a web-based platform that combines gamification, serious games, content editors, and automatic correction strategies to promote science, technology, engineering, and mathematics (STEM) among secondary school students. TAECon allows content creators to create challenges, enigmas, and sessions through a web interface while content consumers (players/students) play the game on a Unity-based client. Boada et al. [50] have adopted a centralized client-server approach, in which content is stored in a database and a web server, and users (admins, content creators, and content consumers) log in to access the game sessions. The downside of such an approach is that the content creators still require the support of an IT team to implement, deploy, and maintain it.

Chabbi et al. [51] proposed a game software architecture to develop configurable serious games for inhibitory and interface control. As a proof of concept, a gamified stop signal task (SST) was implemented as 2D and VR games. The proposed architecture comprises three main components: (i) game scenario configurator, which allows the content creator to define parameters such as the total duration of the game session and desired frequency for the appearance of the stop signal; (ii) scenario-based game builder, which consumes the output of the configurator and instantiates platform-independent components such as "SceneManager" (responsible for loading the 3D assets); (iii) game mode generator, which generates platform-dependent components (e.g., menus, user input handling). The usability of the proof of concept games was assessed with the user experience questionnaire (UEQ), and both modes achieved positive scores. Unfortunately Chabbi et al. did not explore how the game scenario configurator component was implemented, or how flexible or customizable the configurator is.

Lastly, Wang et al. [52] proposed an educational game framework called MEMORABLE (Multi-playEr custoMisable seriOus Game fRAmework for cyBer-security LEarning) that allows instructors to customize games' learning content in the context of cyber-security. Wang et al. state that enabling customization of the game will allow the content to be tailored to a broader audience (e.g., different countries and age groups) in contrast to games with fixed content limited to a single demographic. A prototype multiplayer game called "Cyberpoly" was implemented based on that architecture, following a game logic similar to the "Monopoly" board game, and it was evaluated by professors and students. The feedback gathered showed that the game managed to stimulate and motivate students to learn about cyber-security. However, it also lacked some features, such as greater flexibility in terms of the expansion of topics and content. Furthermore, Wang et al. did not provide information on how the content manager or the game was developed and deployed.

**3. Moirai: The Virtual Serious Games Authoring Platform**

To overcome the one-size-fits-all approach (unmodifiable single scenarios) inherent in the majority of serious games that results in predictable and repetitive game experiences

that, after several sessions, can reduce their effectiveness as teaching tools [13], we present the virtual serious games authoring platform called *Moirai*. It was named after the three Greek goddesses (*Atropos*, *Clotho*, and *Lachesis*) known as "The Fates" or "The Three Fates", that weave the threads of life. *Moirai* is a no-code platform to create or modify serious games that are virtual simulations focusing on decisions and communication skills development. Through *Moirai*, content creators (e.g., educators) can develop a serious game that follows one of the following structures: linear, branched, or pseudo-branched. *Moirai* is comprised of the following two components: (i) editor; (ii) player.

### 3.1. Methodology

The development process followed the agile methodology approach, focusing on an iterative approach and working on delivering small functional prototypes. As defined by Atlassian [53], agile is a group of methodologies that demonstrate a commitment to tight feedback cycles and continuous improvement and that frequent increments let the team gather feedback on each change and integrate it into future releases at a minimal cost. In addition, due to the exploratory nature of this project, agile was a better fit than other methodologies, such as waterfall, which would require features to be completed in linear and sequential phases.

Development of the *Moirai* platform began with the *Person-Centred Serious Games for Mental Health Education* project (presented in Section 4.1). Two development streams occurred simultaneously: the content stream (responsible for creating the narrative and dialogue script) and the tech stream (responsible for creating the *Moirai* player and editor); this paper will focus on the tech stream. Since both streams were codependent (e.g., the game stream had to support the narrative designed by the content stream and provide feedback on feasibility), there was continuous communication between the teams.

The game stream followed a two-week sprint cycle, ending with meetings with members of the content stream and subject matter experts (SMEs), whereby the project requirements, scope and features were revised, and feedback was provided on the current prototypes (see Figure 2), until reaching the final prototype that was deployed for user study sessions.

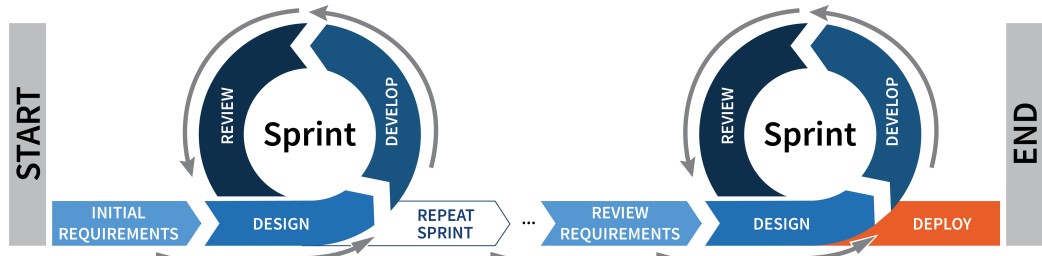

**Figure 2.** *Moirai* development cycle. Adapted from: https://hive.com/blog/what-is-agile-project-management-methodology/ (accessed on 12 November 2022).

### 3.2. The Moirai Editor

The *Moirai* editor allows content creators (e.g., healthcare educators) to create scenarios (and modify them later, if necessary) as needed simply and intuitively, even if the user has a limited (if any) game development/computer science background, through a no-code block-based user interface.

The *Moirai* editor is a single-page web application based on Vue.js (Vue.js—The Progressive JavaScript Framework: https://vuejs.org/, accessed on 23 September 2022), a Javascript framework for developing user interfaces, and BaklavaJS (Newcat/Baklavajs— Graph / node editor in the browser using VueJS: https://github.com/newcat/baklavajs, accessed on 23 September 2022), a graph/node editor for the web. This ensures that the *Moirai* Editor is a proper native web application that can be easily extended in the future to support more features, as required. Both Vue.js and BaklavaJS are open-source projects under the MIT license. *Moirai* will be released as an open-source project under the GPLv3

license (The GNU General Public License v3.0—GNU Project—Free Software Foundation: https://www.gnu.org/licenses/gpl-3.0.en.html, accessed on 23 September 2022) on a public GitHub repository (andreibosco/moirai—Moirai—A No-Code Virtual Serious Game Authoring Platform: https://www.github.com/andreibosco/moirai/, accessed on 2 October 2022).

### 3.2.1. User Interface

The *Moirai* editor's diagram-based user interface (UI) is based on interconnected nodes (see Figure 3), each representing an element in the scenario (e.g., starting point, and dialogue). Detailed information and figures about the available nodes will be provided further ahead in this section. This UI paradigm has been adopted by various software in many fields, including:

- 3D creation: Blender by Blender Foundation (Blender Foundation—Blender 3.0: https://www.blender.org/download/releases/3-0/, accessed on 12 October 2022).
- Game engine: Unreal by Epic Games (Epic Games—Blueprints Quick Start Guide: https://docs.unrealengine.com/5.0/en-US/quick-start-guide-for-blueprints-visual-scripting-in-unreal-engine/, accessed on 12 October 2022).
- Video compositing: Fusion by Blackmagic Design (Blackmagic Design—Fusion 18: https://www.blackmagicdesign.com/products/fusion, accessed on 13 October 2022).
- Machine learning: Azure machine learning designer by Microsoft (Tutorial: Designer—train a no-code regression model—Azure Machine Learning: https://learn.microsoft.com/en-us/azure/machine-learning/tutorial-designer-automobile-price-train-score, accessed on 13 October 2022).

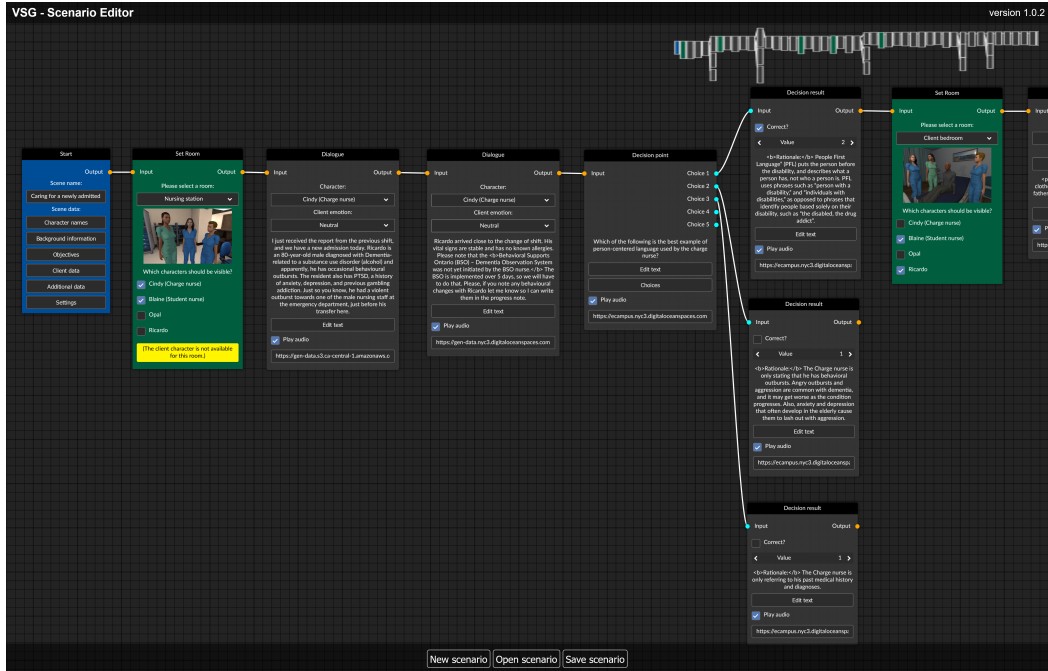

**Figure 3.** Zoomed out view of the *Moirai* editor UI showing the beginning of a dialogue tree with interconnected nodes.

The UI of the *Moirai* Editor is comprised of three main areas (see Figure 4): (i) header (shows the editor name and version), (ii) work area (the main portion of the editor, in which nodes are organized), and (iii) toolbar (contains buttons to create, open, and save scenarios). Within the work area, one overlay is always visible: an interactive minimap, which shows the whole node structure and can be used to navigate quickly throughout the work area.

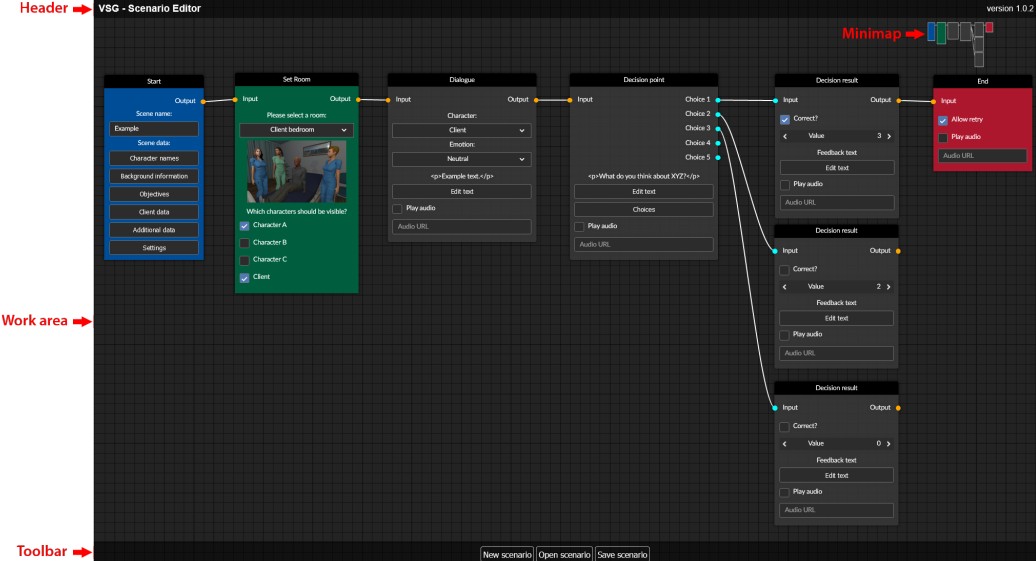

**Figure 4.** Moirai Editor user interface.

To prevent the UI and nodes from becoming too cluttered, some parameters (such as "Additional Data" in the Start Node) are represented as buttons, which activate a sidebar when clicked (see Figure 5). The sidebar appears on the right portion of the screen and does not prevent the content creator from interacting and manipulating the scenario nodes. The sidebar's content varies according to the parameter, but all follow the same structure: name of the node, name of the parameter, brief instructions, and parameter fields (see Figure 6).

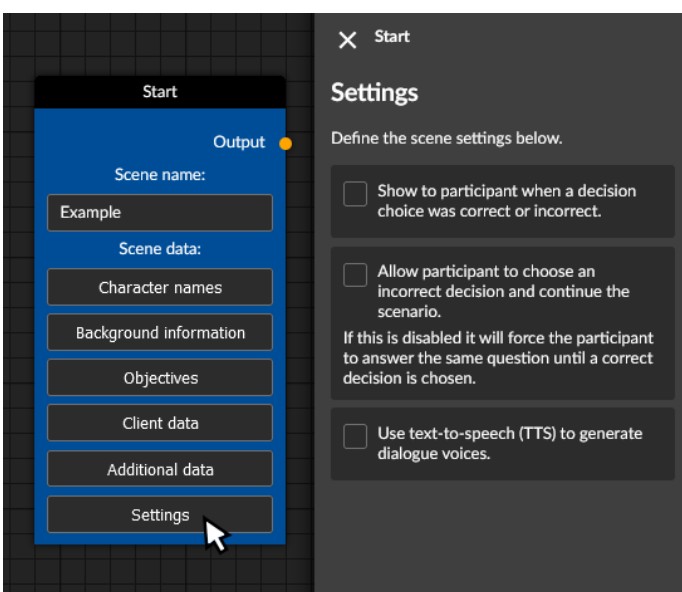

**Figure 5.** Sidebar activated when a user clicks on the "Settings" button of the Start Node.

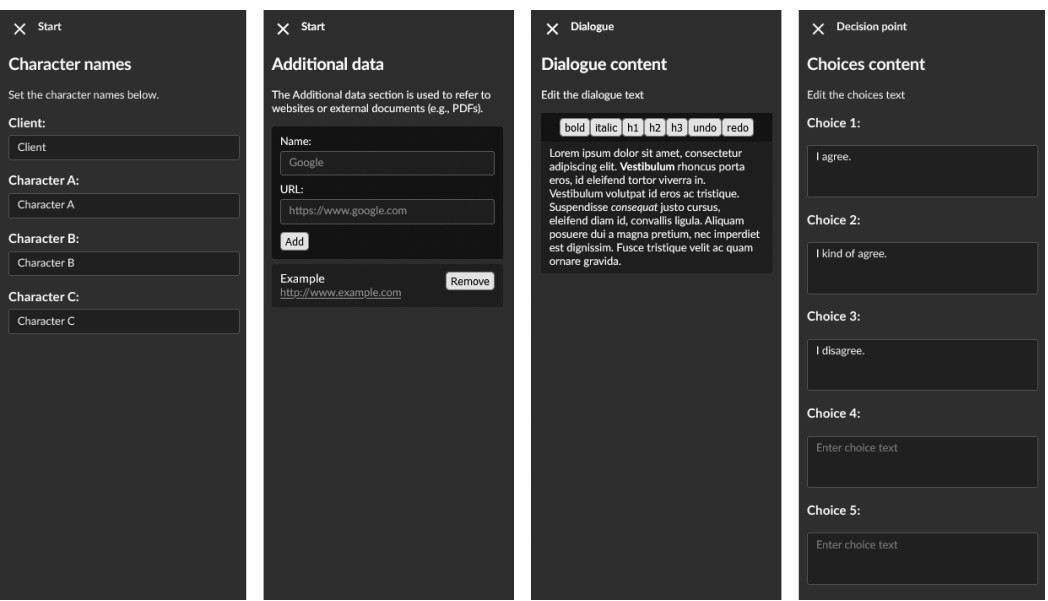

**Figure 6.** Four variations of the sidebar according to the node parameter.

Lastly, to help the content creator to differentiate nodes quickly, they have been color-coded into the following (see Figure 7):

- **Blue:** Marks the starting point of the game. Only one can exist. Nodes: Start.
- **Red:** Marks an ending point of the game. Multiple end points can exist. Nodes: End.
- **Grey:** Identifies nodes that present information or choices to the learner. Nodes: Dialogue, Dialogue with choices, Decision point, Decision result.
- **Green:** Marks nodes that cause a modification to the scenario. Nodes: Set room.

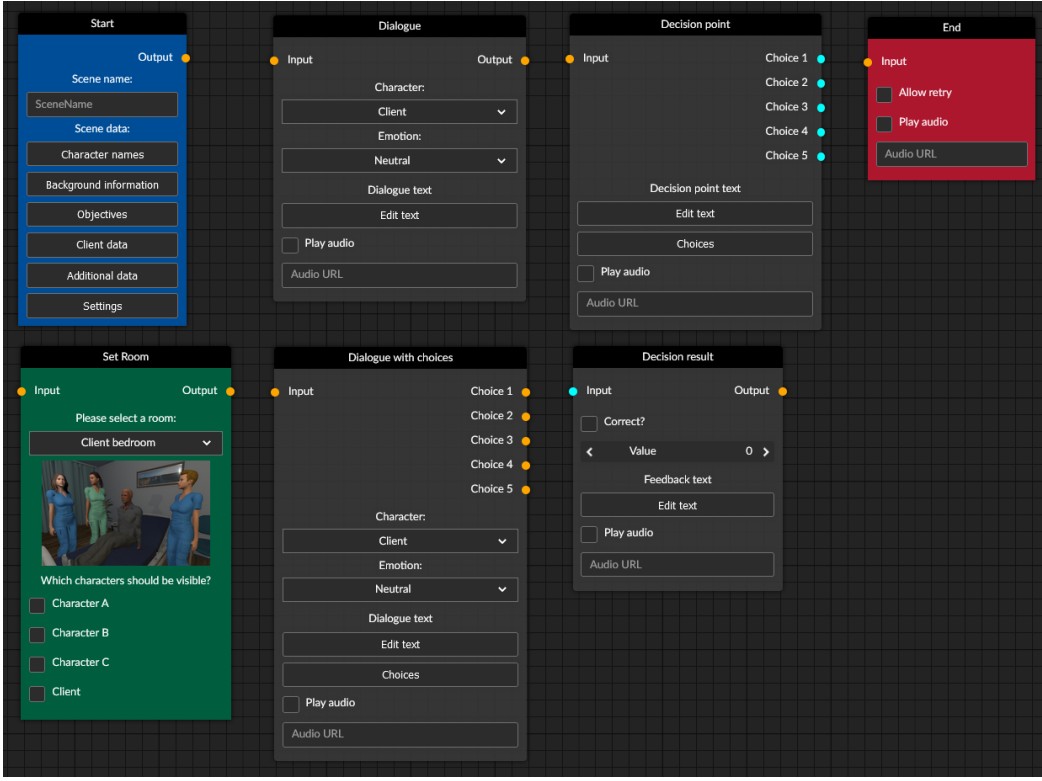

**Figure 7.** All available node types color-coded.

Finally, the content generated in the scenario editor can be saved as a JSON (JavaScript Object Notation) file, which is text-based and human-readable. The JSON file can be opened by the *Moirai* player (and fully experienced as a game), by the *Moirai* editor for further changes or updates, or it could be opened in any text editor and modified manually.

If the end-user refreshes the web page and there is any content in the scenario editor, a warning appears on the screen requesting confirmation to leave the page. This feature provides the content creator confidence that their work is not deleted accidentally. In addition, dialogues with text content support rich text formatting (e.g., bold, italic), and a formatting toolbar is shown in text fields.

### 3.2.2. Nodes

Each node can have input and output interfaces that are connected and disconnected by dragging and dropping. These connected interfaces are read left to right and they determine the flow that the learner will follow during the game, collectively resulting in a dialogue tree with the interactions between the user and the non-playable characters (NPCs) (see Figure 3).

Currently, there are two types of interfaces (represented by the colors orange and cyan, see Figure 8), that visually identify which inputs and outputs can be connected (e.g., a decision point output can only be connected to a decision result input). Additionally, each node includes parameters that define the node's functionality and content. Table 1 lists the current nodes available and their respective components (fields and connections).

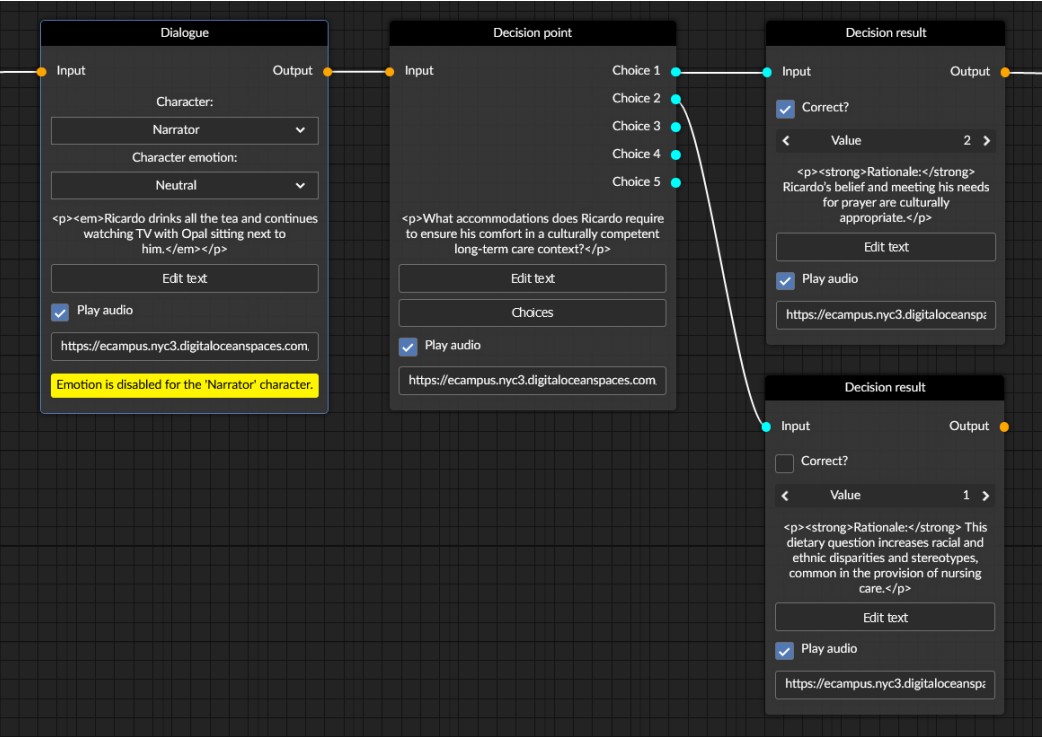

**Figure 8.** Moirai editor nodes with different interface types.

**Table 1.** Available nodes and components

| Node Name | Parameters | Connections |
|---|---|---|
| Start | - Scene name (text).<br>- Character names (text).<br>- Background information (rich text).<br>- Objectives (rich text).<br>- Client data (rich text).<br>- Additional data (list of websites' names and URLs).<br>- Settings (booleans). | - Output (orange). |
| End | - Allow retry (boolean).<br>- Play audio (boolean).<br>- Audio URL (text). | - Input (orange). |
| Dialogue | - Character (list).<br>- Client emotion (list).<br>- Dialogue (rich text).<br>- Play audio (boolean).<br>- Audio URL (text). | - Input (orange).<br>- Output (orange). |
| Dialogue with choices | - Character (list).<br>- Client emotion (list).<br>- Dialogue (rich text).<br>- Play audio (boolean).<br>- Audio URL (text). | - Input (orange).<br>- Output: Choice 1 (orange).<br>- Output: Choice 2 (orange).<br>- Output: Choice 3 (orange).<br>- Output: Choice 4 (orange).<br>- Output: Choice 5 (orange). |
| Decision point | - Dialogue (rich text).<br>- Play audio (boolean).<br>- Audio URL (text). | - Input (orange).<br>- Output (cyan). |
| Decision result | - Correct? (boolean).<br>- Value (integer).<br>- Dialogue (rich text).<br>- Play audio (boolean).<br>- Audio URL (text). | - Input (cyan).<br>- Output (orange). |

Every scenario must begin with a "Start" node (mandatory and unique), which determines the starting point of the game and also allows the content creator to set the initial information that will be relayed to the learner (complete list available in Table 1). The content creator can use the "Additional Data" parameter to define links to external content, such as documents, videos, or other websites. Furthermore, the "Settings" parameter allows the content creator to determine some settings that affect the entire game, such as if the learner should be informed whether a decision chosen was correct or incorrect, or whether the game should use a text-to-speech (TTS) system to generate the dialogue voices automatically.

Most of the text-based parameters in all nodes (except for "Scene name" and "Audio URL") support using rich text formatting (bold, italic, header levels) to present information to the learner.

The "Set Room" node allows the content creator to define the location where the scenario is taking place from a list of predefined rooms and select which characters will be visible and take part in the dialogue (character models and positions are also predefined for each room). When the *Moirai* player detects a "Set Room" node, it executes a fade-out and fade-in effect to transition between the rooms and reposition the characters. On the *Moirai* editor, the "Set Room" node also shows a preview of the currently selected room, allowing the content creator to see the room's layout and the characters' positions (see Figure 9).

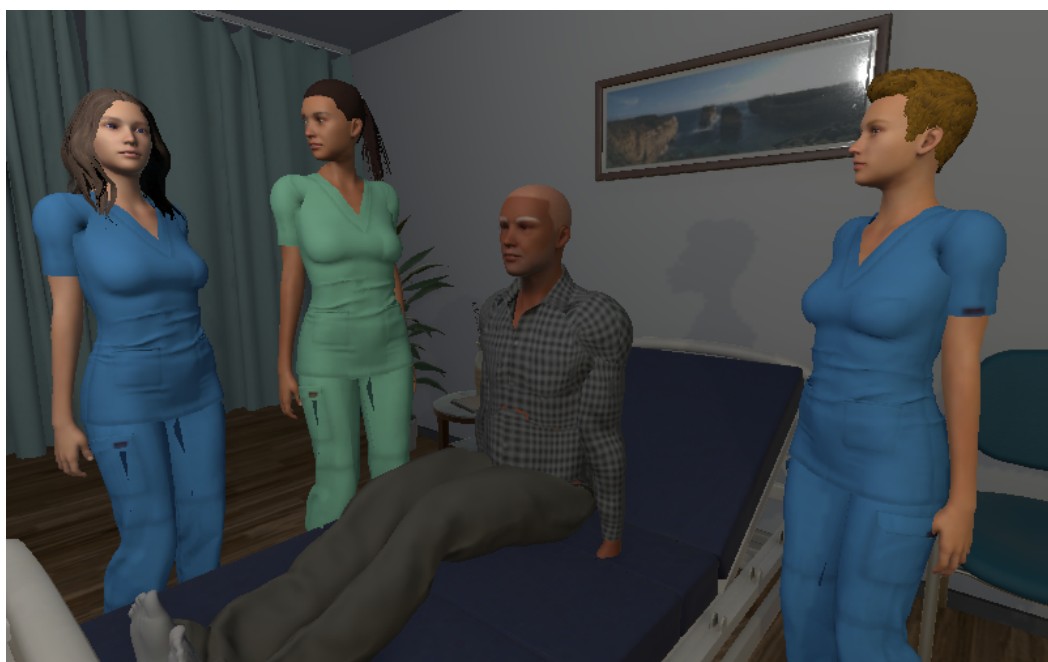

**Figure 9.** Example of a room preview (client bedroom), showing three healthcare providers and a client.

The widely used nodes are the ones that relay dialogue and decision choices to the learner, which are "Dialogue," "Dialogue with choices," "Decision point," and "Decision result." The "Dialogue" and "Dialogue with choices" nodes have the same basic structure: the character relaying the information, its emotion (a predefined list that will control the character's animation), the dialogue text, a checkbox (Boolean) field to determine if audio should be played, and a text field for the audio URL. The difference between them is that the "Dialogue with choices" has five outputs and one additional button: choices. This button allows the content creator to set the text of up to five choices, and each of these choices will follow the path determined by their respective outputs.

The "Decision point" node is similar to the "Dialogue with choices": it also has five output nodes and a button to set the content of the choices, but it does not have any parameters related to characters or emotions. The system narrator will play this node, and its outputs can only be connected to "Decision result" nodes. The "Decision result" nodes can be defined as correct or incorrect, have an integer value, feedback text (to explain to the learner why that specific choice should or should not be made), and also audio playback (see Figure 8).

Lastly, as described previously, most nodes have a field to determine if audio should be played and a text field to set the audio file URL. This permits the content creator to create voice-overs of the dialogue and messages. However, another option is enabling the "Use text-to-speech (TTS)" option in the "Start" node, which will inform the *Moirai* player to use the Google Cloud TTS system to generate the audio files dynamically.

### 3.3. The Moirai Player

The second component of the proposed authoring platform, the *Moirai* player, was developed using Unity (Unity Technologies—Unity Real-Time Development Platform | 3D, 2D VR & AR Engine: https://unity.com/, accessed on 23 September 2022), a cross-platform game engine widely adopted by various industries, such as film, architecture, automotive, and engineering. Using Unity ensured the availability of assets that allowed a faster iteration pace for prototypes' development while also allowing for games to be developed for WebGL, thus ensuring anyone can access and use the *Moirai* player via a regular web browser without needing to download and install it on their device. Furthermore, to prevent users from having to download large files over the web, we reduce texture file sizes

(highest resolution is 512 × 512 pixels), adopt models with low polygon count (making manual adjustments when necessary), and constantly monitor build reports to identify unused components that can be removed and areas that need improvement. Currently, the *Moirai* player file size is around 84 megabytes (including two 3D scenarios, furniture assets, and character). Additionally, we are working on streaming assets as needed, meaning that data would be loaded dynamically instead of preloading all assets when starting the game.

To ensure compatibility of the *Moirai* player with existing LMS, we have adopted the SCORM standard. SCORM is the de facto industry standard for eLearning interoperability (SCORM Explained 101—One Minute SCORM Overview: https://scorm.com/scorm-explained/one-minute-scorm-overview/, accessed on 14 October 2022), and it determines how online learning content (such as the *Moirai* player) and the LMS communicate with each other. Therefore, the *Moirai* player can be imported into various LMSs such as Moodle (Using SCORM—MoodleDocs: https://docs.moodle.org/400/en/Using_SCORM, accessed on 14 October 2022), Canvas, Absorb (Absorb LMS—SCORM Compliant LMS: https://www.absorblms.com/features/scorm-compliant-lms, accessed on 14 October 2022), or the GEN [54].

### 3.3.1. Structure

The *Moirai* Player comprises four main components (see Figure 10): controllers, characters, rooms, and interface.

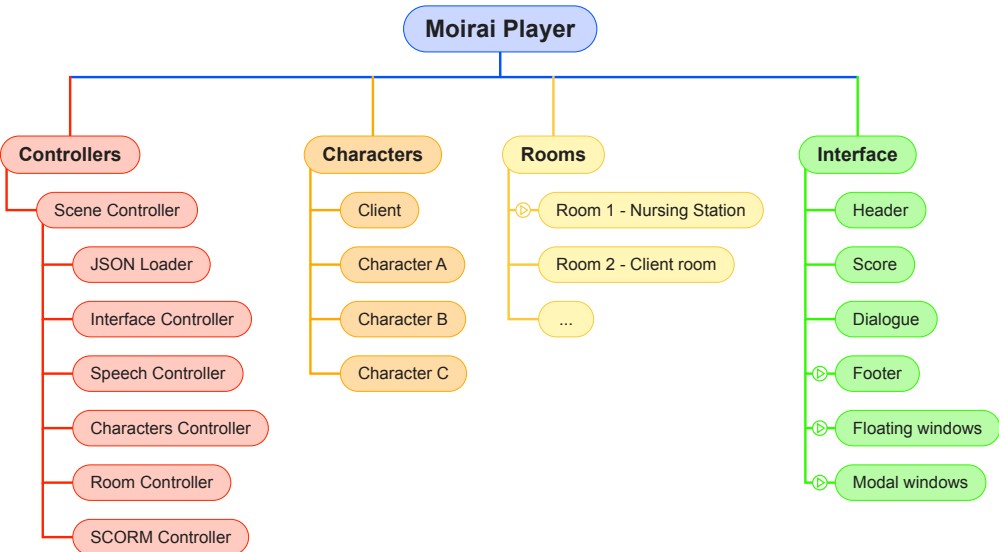

**Figure 10.** Structure of the Moirai player components.

The controllers are components responsible for the logic and the game's functionalities. When the *Moirai* player is started, the "Scene Controler" acts as the manager, handling the execution of the game and relaying calls to the other controllers, briefly described below:

- **JSON Loader:** handles loading of the JSON file data.
- **Interface Controller:** handles all the user interface components.
- **Speech Controller:** if TTS is enabled, it will connect to Google Cloud TTS and convert all text into audio.
- **Characters Controller:** handles character-related data and events (e.g., names, and animation).
- **Room Controller:** handles rooms and transitions between them.
- **SCORM Controller:** handles communication between the player and the LMS, relaying data such as score, status, and time.

The character components are objects (called *GameObjects* within Unity) representing the client and other NPCs in the scenario. These characters are defined using the Unity

Multipurpose Avatar System (UMA) (Umasteeringgroup/UMA—Unity Multipurpose Avatar: https://github.com/umasteeringgroup/UMA, accessed on 14 October 2022), an open-source character creation and modification system. The UMA allows us to quickly customize the characters according to the needs of the scenarios. Morever, to help enhance the realism while keeping everything procedural, we have adopted the SALSA (Simple Automated LipSync Approximation) LipSync suite (Crazy Minnow Studio—SALSA Lip-Sync, EmoteR, and Eyes for Unity: https://crazyminnowstudio.com/unity-3d/lip-sync-SALSA/, accessed on 14 October 2022), to dynamically animate the characters' mouths according to the audio files provided either by the content creator or by the TTS system. That suite also allows us to dynamically animate the characters' eyes, eyelids (making them blink), and heads, allowing us to make the characters look at specific positions or at the other characters. Additionally, we have used and modified a set of royalty-free animations by Mixamo (Adobe—Mixamo: https://www.mixamo.com/, accessed on 14 October 2022) to convey the emotions defined by the content creator in the *Moirai* editor.

The room components are composed of a group of objects that represent the 3D objects of the scenario to be represented, a camera object that represents the player's point of view, and references of where the character components should be positioned within the room.

Lastly, the interface components represent all the user interface elements, which are presented in Section 3.3.2.

### 3.3.2. User Interface

The *Moirai* player loads the scenarios created in the *Moirai* editor and provides the learner with a simple interface to go through ("play") the scenario. When playing for the first time, a brief introduction guide is presented with textual information and images, explaining how to use the learner (see Figure 11). Afterwards, the learner is presented with the information defined in the "Start Node," such as background information and additional data (see Figures 12 and 13). That ensures that the learner is aware of the information necessary to grasp the scenario being played. Subsequently, the scenario starts to be played following the structure set in the *Moirai* editor.

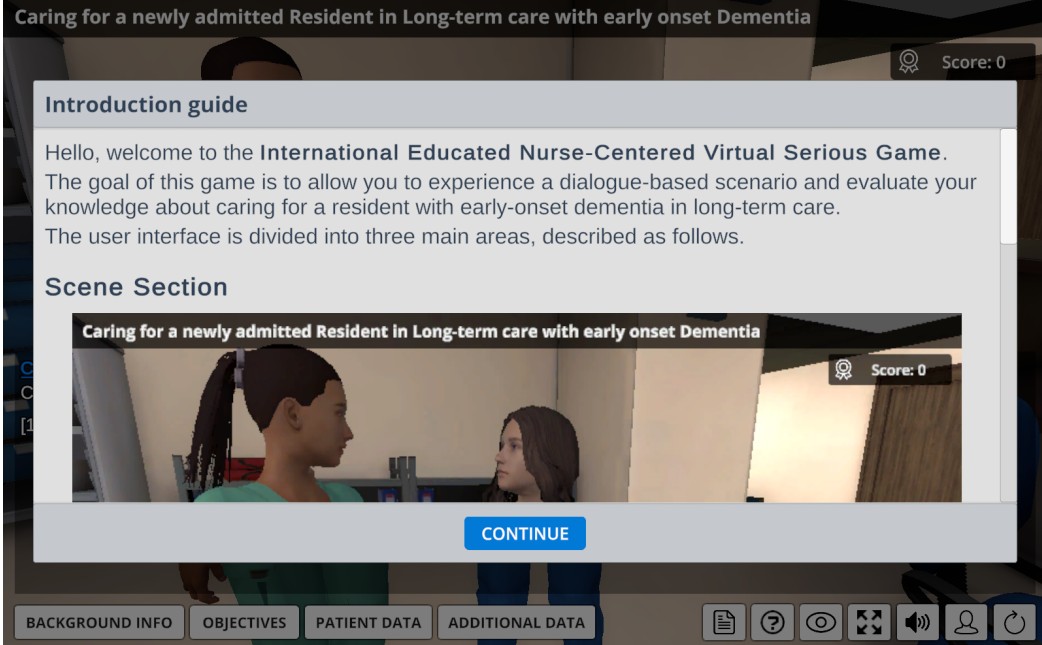

**Figure 11.** Introduction guide that is shown upon the first time playing the resulting game.

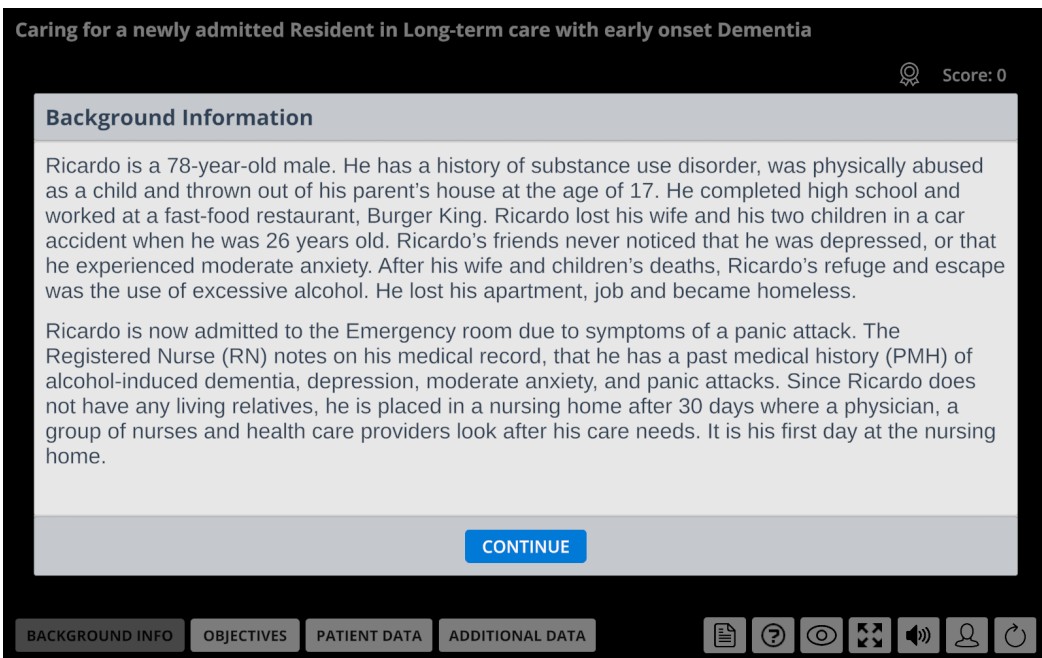

**Figure 12.** Background information window of one of the scenarios developed.

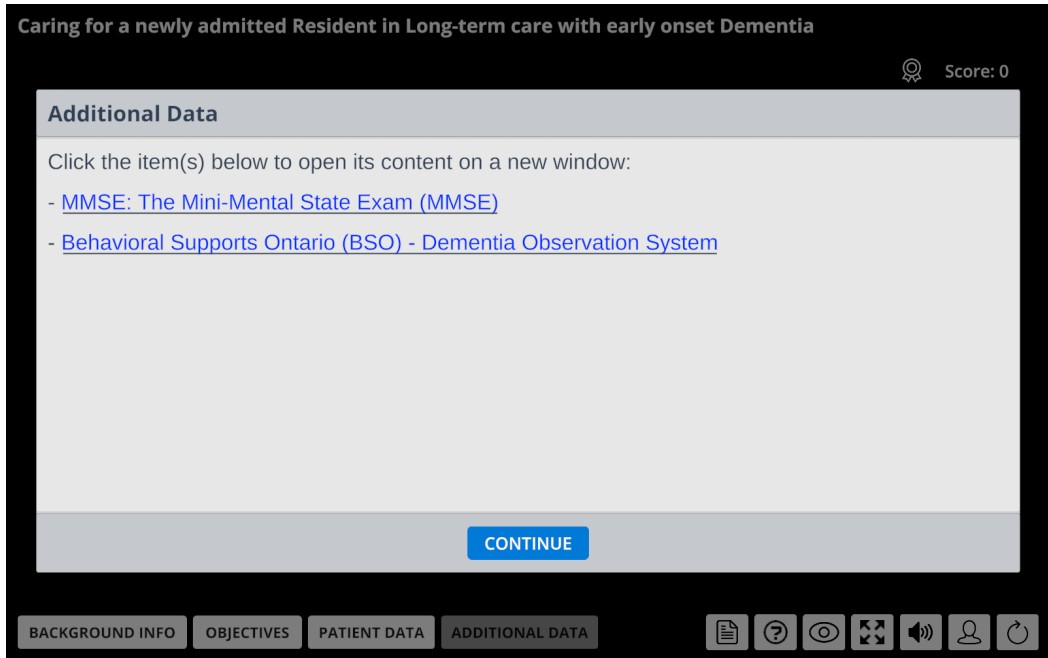

**Figure 13.** Additional information window listing two links for documents on the web.

After going through the initial scenario information, the learner is presented to the main UI of the *Moirai* player (see Figure 14). Similar to the *Moirai* editor, it is composed of three main areas: (i) the header (listing the name of the scene, as defined in the "Start Node"); (ii) the scene area (which makes for the main portion of the screen, showing the 3D environment and characters); and (iii) the footer (which contains a list of buttons). Within the scene area, there are two components: (i) the score counter and (ii) the dialogue area (where all characters' dialogue, narrator's feedback, and choices will be displayed). Both have a semi-transparent background to prevent obscuring the 3D environment and characters and provide a greater point-of-view. The learner will experience the scenario from either a first-person or third-person point of view, depending on how the room was designed.

At any moment, the learner can review any of the introductory information by using the buttons located in the left portion of the footer area (see Figure 14). Additionally, on the right portion of the footer area, there are buttons to allow the learner to see a textual log of the scenario up to that moment (including the choices made), review the introduction guide again, control the virtual camera and look around the current room, set the *Moirai* player fullscreen, toggle the audio playback, open the credits screen, and restart the game.

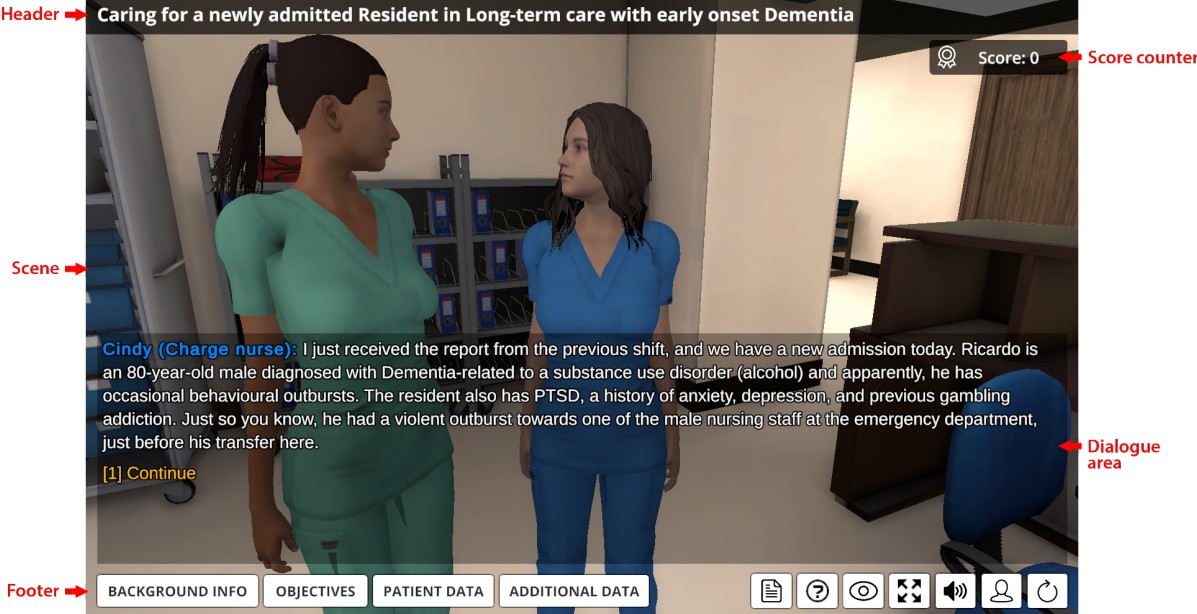

**Figure 14.** Moirai player user interface.

Currently, the rooms developed for the *Moirai* can contain up to five characters: a client, three NPCs, and a narrator. The client and NPCs are physically in the scenario, while the narrator is a disembodied character that can provide additional information and explanations to the learner. Based on the settings defined by the content creator in the *Moirai* editor, the NPCs will talk to the learner and include animated emotional responses (e.g., anger, agitation, sadness) depending on the learner's choice. In addition, as mentioned in Section 3.2, all dialogue text can be played to the learner by automatically converting it to audio using TTS or by using audio URLs defined on the *Moirai* editor.

When the learner reaches the end of the scenario (defined as the "End Node" in the editor), it is presented with a screen presenting the final score percentage obtained, a breakdown of the score (raw score, maximum and minimum score possible), and a trophy badge based on the percentage obtained (see Figure 15). Additionally, the learner can review the final log of the scenario and play again (if the setting was enabled in the "End Node"). The ability to restart the scenario illustrates the value of such technology as learners can practice and master interactions during their own time and with feedback.

Moreover, the *Moirai* player includes keyboard navigation allowing the learner to navigate between the user interface elements using the keyboard (e.g., moving up and down the dialogue choices and buttons using the keyboard arrow keys).

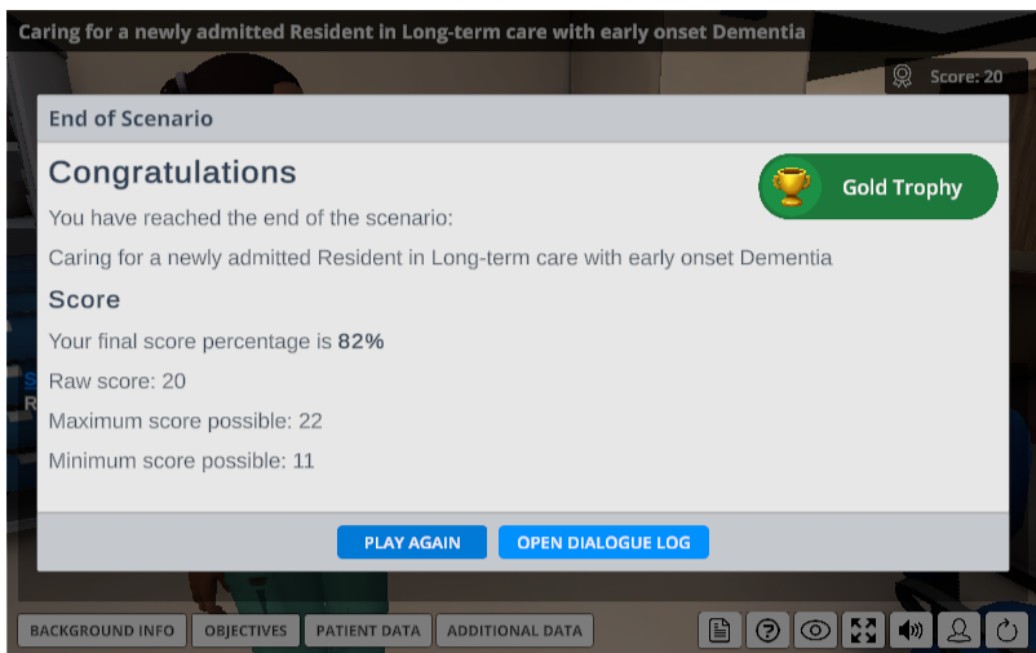

**Figure 15.** End of scenario screen, showing final score percentage, score breakdown, and trophy badge.

## 4. Case Studies

During the development of *Moirai*, two case studies were formulated and evaluated throughout 2019 and 2020, within the mental health education and internationally educated nurses learning domains.

### 4.1. Person-Centred Serious Games for Mental Health Education

The development of this case study started in 2019, prior to the COVID-19 pandemic, and it addressed two problems: (i) the need to better understand the lived experiences of persons with mental illness to inform the design of serious games geared for undergraduate students, and (ii) the need for more accessible and interactive VSGs for student learning prior to attending mental health community placements [55].

#### 4.1.1. Context

Every year, one in five Canadians will experience or be affected by persons living with mental illness [56,57]. Physical and mental health are linked, so students in programs such as nursing, psychology, social work and front-line workers, i.e., paramedics will ultimately be exposed to persons with mental health issues [8]. However, mental health education varies across Canada, and there is a gap in providing a consistent approach to undergraduate education, especially in nursing [58]. A survey of 30 Canadian undergraduate schools of nursing found that 20% of programs did not offer a mental health course and mental health clinical placement. Moreover, mental health theory and community placements in nursing have continued to decline [8]. To improve the care of persons living with mental illness, nursing students require quality education that includes theoretical courses and placement opportunities. Nursing placements in mental health settings have been difficult to obtain with higher student enrolment rates, staffing shortages, and higher patient acuity [59,60]. Therefore, novel educational strategies are required so that students acquire the necessary knowledge, skills, and attitudes to competently care for persons living with mental illness.

The current literature also confirms that most undergraduate students have negative attitudes due to reduced engagement in community settings and a lack of consideration for mental health careers [61–63]. Students who have secured mental health placements still experience knowledge and attitudinal gaps. The effectiveness of a student's placement

experience can also be dependent on preceptor and staff support; moreover, student evaluation can be challenging as the presence of an instructor during a student–client interaction can hinder a therapeutic relationship [59,64,65].

Regardless of the lack of placement opportunities and associated challenges, alternative educational approaches can contribute to undergraduate student learning in the mental healthcare domain. For example, one approach to dealing with mental health placement challenges is using VSGs informed by the lived experience of persons with mental illness.

### 4.1.2. Scenario

In 2019 we collected narratives from individuals who live with mental illness and accessed mental health services using qualitative methods [55]. We used the findings to co-design a VSG representative of their lived experience and support for shared decision-making [17]. In the resulting scenario, the player (nursing students) must conduct a mental health assessment on the client's character (see Figure 16). The error-tolerant environment allowed students to make mistakes and learn from those errors while minimizing patient risk.

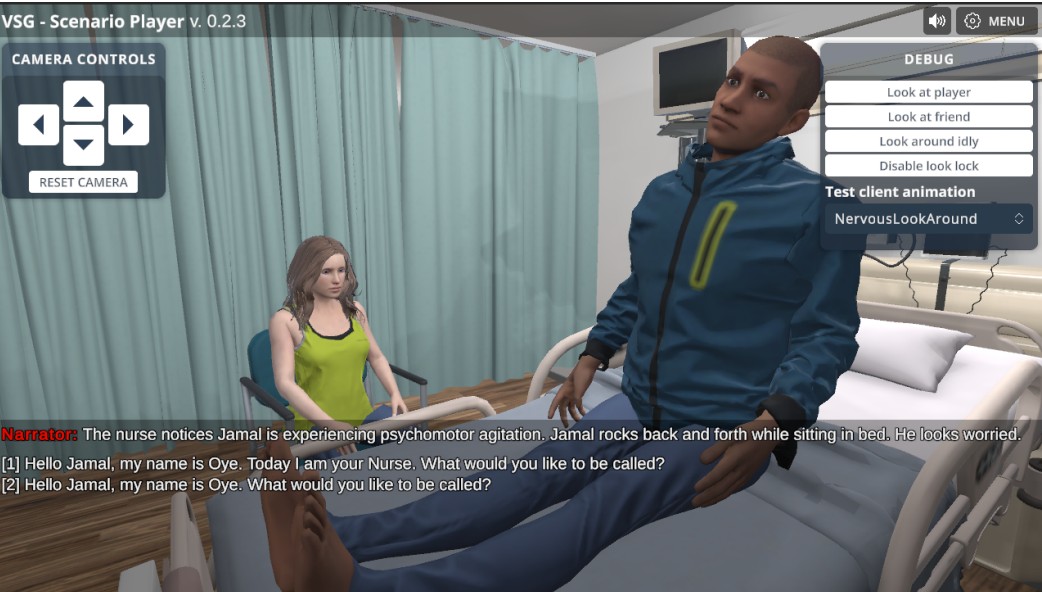

**Figure 16.** Earlier version of the Moirai player, showing a client sitting in a hospital bed next to a friend sitting on a chair.

### 4.1.3. User Study

To evaluate the usability of the *Moirai* player and the scenario developed for this study, the system usability scale (SUS) [66] was applied. The SUS is a questionnaire composed of 10 items with responses following a 5-point Likert scale (from "strongly disagree" to "strongly agree"). The items alternate between positive and negative statements to encourage respondents to read each statement and make an effort to decide whether they agree or disagree [67]. The SUS yields a single number representing the overall usability, and the scores of the individual items are not meaningful on their own [66].

It is also important to note that although the SUS scores range from 0 to 100, they are not percentages, and should be considered in terms of their percentile ranking [68]. A SUS score above 68 is considered above average, and anything below 68 is below average [68].

The SUS was applied to 18 participants (n = 18), of which 15 were nursing students and three were nursing faculty. The final SUS score obtained was 85, implying that the *Moirai* player achieved a good degree of usability [55].

### 4.2. International Educated Nurse-Centred Virtual Serious Game

4.2.1. Context

High-income countries (HICs) such as Canada have remedied shortages in the nursing workforce by hiring internationally educated nurses (IENs). However, Canada has demonstrated an inadequate response to the learning needs of IENs [10]. IENs have unfair access to educational activities, leading to feelings of invisibility and marginalization [11,69,70]. Many IENs do not pass the standardized examination and requirements set by regulatory bodies such as the College of Nurses in Ontario (CNO), partly because they do not provide open access to educational resources [69,71–73]. Such educational resources must be virtual and flexible to accommodate the work-life of IENs. Although post-secondary IEN bridging programs are available, they lack standardization and often fall short of quality education. The "deskilling process" also results in IENs working at lower-level jobs for long periods because of contradictory nursing education programs [9,71].

4.2.2. Scenario

A needs analysis and a needs assessment survey were performed in 2021 to guide the development of the scenario for this study case [18]. The resulting scenario includes an elderly male patient with dementia and a history of depression who was admitted to a long-term care facility. The learner takes the role of a nurse who must complete a mental health assessment, provide a gentle persuasive approach, administer medications, and provide client-centered care (see Figure 17).

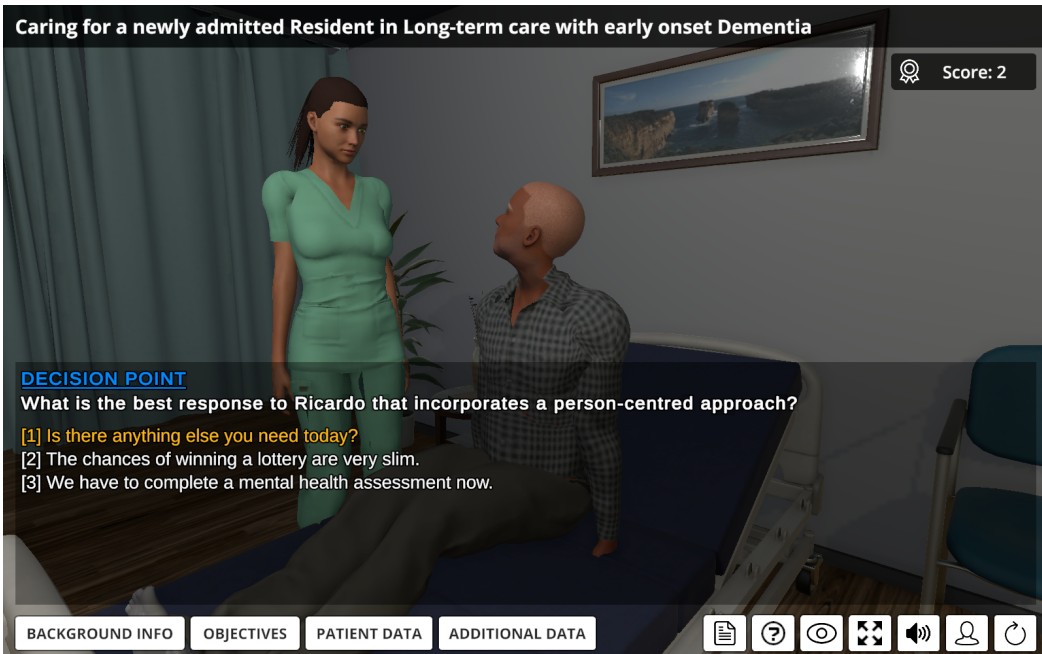

**Figure 17.** Player represented by a nurse interacting with an elderly client and having to choose the best response in a decision point.

4.2.3. User Study

Similar to the previous study, the SUS questionnaire was applied to evaluate the usability of the developed *Moirai* player and scenario for this study. The SUS was applied over email to 14 participants (n = 14), of which 11 were nursing students and 3 were nursing faculty. Again, as in the previous study, the SUS score obtained was approximately 85, or in the percentile range of 86-88 [18], which implies that we have achieved above-average usability.

## 5. Challenges and Limitations

We recognize that developing a general tool to empower users with limited technical knowledge is challenging and that some restrictions will be necessary. Our main challenge is finding the middle ground between ease of use versus functionality. We do not want to make an extremely friendly but technically limited platform that will not be useful for educators. Another considerable challenge is having the web (and, consequently, LMSs) as our target platform. That imples having a broad range of browsers and systems that needs to be tested (e.g., Safari on macOS has a limited implementation of WebGL). Furthermore, file size is also an essential factor since we cannot guarantee that users will have access to a fast internet connection.

Regarding limitations, game developers and 3D artists are currently necessary to design the 3D environments and characters that will be available on the *Moirai* platform for educators and content creators. We intend to expand the *Moirai* Editor to allow the creation of the environments, importing custom 3D objects, and customizing characters. However, the work on those features has not yet begun.

## 6. Conclusions and Ongoing Work

Our work-in-progress research involves the ongoing development of *Moirai*, a virtual serious games authoring platform, that allows educators to develop serious game scenarios using a no-code approach. By providing educators with the tools and training necessary to build their interactive content, we aim to see a proliferation of such content for all domains, including healthcare education and training.

Here we outlined *Moirai* and described two case studies, each of which involved creating serious games for nursing education (one aimed at mental health education and the other aimed at internationally educated nurses) using *Moirai* and evaluating their usability with human participants. The usability of both serious games was evaluated qualitatively using the system usability scale (SUS) questionnaire with practicing nurses and nursing educators, and it achieved above-average usability scores.

Work on *Moirai* is ongoing, and currently, the *Moirai* player is being optimized for smartphones, and we also intend to have a version for virtual reality headsets (such as the Meta Quest). We are also evaluating different TTS systems that provide more natural-sounding voices. Moreover, under evaluation is the need to add more nodes to create a more dynamic scenario (e.g., visual effects node), expanding the functionality of existing nodes (such as setting a time limit to choose an answer), and allowing the content creator to customize how the characters look (e.g., clothing, hair, age, skin tone). The main challenge is to find the balance between feature richness and ease of use.

Although it is unrealistic to expect educators to be highly skilled in game development and programming, one should expect that some would be willing to take the time to learn some basic skills to provide a better learning environment for their students. To help bridge that gap, we will also implement a tutorial to guide new users to the *Moirai* platform, add hints, and a help section. Nevertheless, what is the appropriate trade-off between functionality and ease of use? Or how much time are educators willing to devote to learning how to use such tool if it was available? To this end, we are in the process of designing a survey to gauge the technical skill set and knowledge of healthcare educators, and their willingness to develop interactive software and games (given appropriate development tools). The survey is being developed using the Delphi method [74] to ensure that the survey questions provide a suitable assessment by a panel of experts and to determine whether the questions are appropriately geared for learners' and research goals. Once completed, it will be sent to health educators, and the results will guide the further development of our *Moirai* editor. Lastly, we will also expand our usability studies by evaluating the *Moirai* editor to help us achieve our goal of balancing user-friendliness and functionality.

**Author Contributions:** Conceptualization, A.T., B.K., C.D.S., E.P. and A.D.; methodology, A.T., B.K., C.D.S., E.P. and A.D.; software, A.T.; validation, C.D.S., E.P., A.T., B.K. and A.D.; formal analysis, C.D.S.; investigation, A.T., B.K., and A.D.; writing—original draft preparation, A.T., B.K. and C.D.S.; writing—review and editing, C.D.S., E.P., A.T., B.K. and A.D.; supervision, B.K. and A.D.; project administration, B.K. and A.D.; funding acquisition, C.D.S., E.P., B.K. and A.D. All authors have read and agreed to the published version of the manuscript.

**Funding:** This research was funded by the *Social Sciences and Humanities Research Council of Canada* (SSHRC) in the form of an Insight grant to C. Da Silva, E. Peisachovich, B. Kapralos, and A. Dubrowski, the *Natural Sciences and Engineering Research Council of Canada* (NSERC), in the form of a Discovery grant to B. Kapralos, the *Ontario Tech University Research Excellence Chair* to B. Kapralos, and by the *Government of Ontario* through the eCampus Ontario's support of the Virtual Learning Strategy to C. Da Silva, E. Peisachovich, B. Kapralos, A. Dubrowski, and A. Torres, and through an Ontario Trillium Scholarship to A. Torres.

**Institutional Review Board Statement:** The study was conducted in accordance with the Declaration of Helsinki, and approved by the Ethics Review Board of York University (Certificate Number: 2019-424).

**Informed Consent Statement:** Informed consent was obtained from all subjects involved in the studies.

**Data Availability Statement:** Not applicable.

**Conflicts of Interest:** The authors declare no conflict of interest.

## Abbreviations

The following abbreviations are used in this manuscript:

| | |
|---|---|
| CNO | College of Nurses in Ontario |
| EUD | End-User Development |
| GPE | Graphical PalCom user interface markup language editor |
| HIC | High-Income Countries |
| IEN | Internationally Educated Nurse |
| iVLE | Immersive Virtual Learning Environment |
| JSON | JavaScript Object Notation |
| LMS | Learning Management System |
| MEMORABLE | Multi-playEr custoMisable seriOus Game fRAmework for cyBer-security LEarning |
| NPC | Non-Playable Character |
| RAD | Rapid Application Development |
| SALSA | Simple Automated LipSync Aproximation |
| SCORM | Sharable Content Object Reference Model |
| STEM | Science, Technology, Engineering, and Mathematics |
| SUS | System Usability Scale |
| TTS | Text-To-Speech |
| UEQ | User Experience Questionnaire |
| UI | User Interface |
| UMA | Unity Multipurpose Avatar |
| VSG | Virtual Serious Game |

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
