# Peer review of "Moirai: A No-Code Virtual Serious Game Authoring Platform"

_2813-2084, doi:10.3390/virtualworlds1020009_

Round 1

Reviewer 1 Report

Comments and Suggestions for Authors

Summary: This article articulates the need for serious games that can be quickly adapted by end-users to their specific needs using low- or no-code methods. The authors focus on the authoring platform Morai and its ability to craft serious games for the healthcare sector. The study discusses in detail the Morai platform and provides 2 examples of applications in the healthcare section, more specifically in nursing. Findings from their case studies found that the serious games developed using Morai had a good degree of usability. 

Introduction: Although I appreciate the COVID-19 angle in the introduction, I think there should also be more of an emphasis on the general need for these serious games before the pandemic, and more discussion of how they were used before 2019.  

General comment: I would suggest thinking about the audience of the paper a bit. I see 3 potential audiences: healthcare educators, serious game developers and the very specific audience of those designing games for healthcare (such as the authors). Myself, I come from a background focused on serious games, so when topics such as those in lines 34-44 come up which is wording specific to healthcare professionals, I have a bit of a hard time following. Depending on who the audience is, you might want to clarify some concepts. From the abstract and conclusion, it sounds like you wanted to reach a wider audience of people working in the field of serious games. If I understand this correctly, I would suggest explaining a bit more contexts that are specific to healthcare education. 

General comment: On the same note as above, understanding exactly who would use Morai would be interesting to add. For example, is it only for healthcare? The current examples are for nurses, does it have applications for other healthcare professionals? Would teachers be creating or adapting these games? From your case studies, it sounds like you (the authors) developed the games, not faculty. Would it be realistic for faculty to take the time to develop these (I see you briefly mentioned it in the conclusion)? If not, who would develop the games? These are just some general questions! I don't think all need to be specifically addressed in the text, but I do think that explaining in more detail in the introduction your target audience and intended use of the platform would be important.

Paragraph at lines 48-53: Serious games for training in healthcare have been used for some time now. It would be great to include an example. 

Line 85: the text appears to have been cut off. The description of what comes after Section 3 is missing and should be added. 

Figure 1: I don't personally think the figure is necessary. Your text above explains everything very well.

Section 2.2.: I don't personally think that there's a need for all the definitions you included as lists in this section. I think this could be done more succinctly. You risk losing the reader by having them read all these definitions that are not referred to later in the text. 

Paragraph lines 110-120: I'm a bit confused here, I think the paragraph would benefit from some rewording. The first sentence says that serious games are often "confused" with "virtual simulation". From what you describe later in this paragraph, I'm not sure if you're trying to make the argument that some serious games are indeed virtual simulations or not.

Line 282: I think you should either name the Greek goddesses or remove the reference. It's small, but it just makes the reader stop and think about something else as they search for who those goddesses are (at least it did for me!).

Line 289: Where do the existing scenarios you mention come from? Do you mean earlier versions of a scenario that the healthcare professional created? 

Figures 8-11: I don't personally think it's necessary to include these since it's hard to see details in the digital version of the article. 

Section 4.1.1. and 4.2.1: Great sections to set the context! I would love to see some of that context setting in easy-to-understand language in the introduction of the paper. By this, I mean better identifying the needs of healthcare professionals that Morai could help fill.  

Figure 24: There's a period missing after Oye in option 1 that you might want to fix.

These are small grammar/typos I caught. Make sure to reread it carefully before your last submission. Overall, the article was well written 

- Line 82, it should be "follows"

- Line 91, it should be "in" revenue

- Line 276, should it be "was"instead of "were"?

- Line 367, it should be "the" same basic structure

- Line 397, it should be "LMS" since you defined is as plural earlier on

- Line 428, it should be "the rooms components' "

- Line 452, it should be "from either"

- Line 518, comma missing after scenario

Reviewer 2 Report

Comments and Suggestions for Authors

Moirai: A No-Code Virtual Serious Game Authoring Platform

When the games become very familiar, when the user knows all the scenarios, options, the game often becomes boring, this is evidenced when the user feels the need to change the game or when he waits for the release of a new version with new scenarios. .

This article presents a tool or platform as the authors call it that allows the customization, change, rotation of game scenarios through a user-configurable graphical web interface.

I consider that the research carried out by the authors is very good, the approach and the idea of ​​the work is innovative and the customization possibilities offer good opportunities to the user.

I found the article easy to read, a good positioning of the proposal against related works, well referenced. Through the development of the platform, the authors demonstrate knowledge and application of the latest technology, they offer a customizable web platform with open technology, something that is also remarkable.

On the other hand, although the concept offered by the platform sounds interesting, I also find that it is limited, for example, to pre-designed scenarios for the purpose of personalization, for example, what happens if I want to design a simulation game not for nursing training but for training of executives, will once again depend on the illustrator and perhaps even the programmer?.

I also think that although the user without knowledge can customize the experience, it requires prior preparation, it is not as easy as it is claimed, otherwise the usability study of this module should be presented. It is important to point out that these limitations do not detract from the work at all, they are simply limitations of the concept that can surely be improved with much more work.

Little suggestions

I think it is necessary to expand the usability evaluation section of the proposal, especially to evaluate the customization scenario and see the results.

Although the authors comment on the development of the tool, I think that it could go deeper into its design stage through a class model, for example, the methodological part.

 The work claims to have an MIT license, I could not find the address or domain of the platform, is it available on the web? I think the work would have the desired impact if it is disseminated, used and shared, that would give it more value to the authors' work.

In general, the figures are good, however figure 7 requires an enlargement of the area in which attention is required, for example as in figure 13. Check that they are all legible.

The conclusions must be improved, the first paragraph is repetitive where once again I feel an introduction.

After these small suggestions, I think the work can be published, congratulations to the authors.

Reviewer 3 Report

Comments and Suggestions for Authors The authors have reported many significant details of what they have achieved.
The authors describe a software for the rapid customisation of virtual environments to be exploited in the medical sector
What you get is a serious game, in this case a video game environment that aims at education and not entertainment.
The article is well written and the sections are correctly elaborated.
The literature search is thorough and extensive.
I might suggest reducing the number of citations, 94 might be too many.
It must be said that many of the items you cite are actually websites and not articles or books, in which case it would be worth considering whether to include them as footnotes or in some other way
Please note that Unity3D has changed its name and is now called just Unity, without the term 3D.
Figure 7 is difficult to read, I suggest reducing the elements shown to enlarge the text
In my opinion, a small section describing how you have optimised the performance of your application would have been appreciated.
The authors on line 292 state that the project is released under the MIT licence, it would be interesting to also include the link to the repository (e.g. GitHub) so that the code can be viewed
These kinds of projects, if made open source, can help the scientific community to grow and improve
